# Decellularized Umbilical Cord as a Scaffold to Support Healing of Full-Thickness Wounds

**DOI:** 10.3390/biomimetics9070405

**Published:** 2024-07-03

**Authors:** Albina A. Kondratenko, Dmitry V. Tovpeko, Daniil A. Volov, Lidia I. Kalyuzhnaya, Vladimir E. Chernov, Ruslan I. Glushakov, Maria Y. Sirotkina, Dmitry A. Zemlyanoy, Natalya B. Bildyug, Sergey V. Chebotarev, Elga I. Alexander-Sinclair, Alexey V. Nashchekin, Aleksandra D. Belova, Alexey M. Grigoriev, Ludmila A. Kirsanova, Yulia B. Basok, Victor I. Sevastianov

**Affiliations:** 1Research Department of Biomedical Research of the Research Center, S.M. Kirov Military Medical Academy, 194044 St. Petersburg, Russia; 2Department of Histology and Embryology, St. Petersburg State Pediatric Medical University, 194100 St. Petersburg, Russia; 3Department of Pharmacology with a Course of Clinical Pharmacology and Pharmacoeconomics, St. Petersburg State Pediatric Medical University, 194100 St. Petersburg, Russia; 4Cellular biotechnology Centre for Cell Technology (CCT), Institute of Cytology of the Russian Academy of Sciences, 194064 St. Petersburg, Russianbildyug@gmail.com (N.B.B.);; 5Department of General Hygiene, St. Petersburg State Pediatric Medical University, 194100 St. Petersburg, Russia; 6Laboratory “Characterization of Materials and Structures of Solid State Electronics”, Ioffe Institute, 194021 St. Petersburg, Russia; 7Department for Biomedical Technologies and Tissue Engineering. Shumakov National Medical Research Center of Transplantology and Artificial Organs, 123182 Moscow, Russia

**Keywords:** umbilical cord, decellularization, scaffold, wound healing, tissue engineering, regenerative medicine

## Abstract

The umbilical cord is a material that enhances regeneration and is devoid of age-related changes in the extracellular matrix (ECM). The aim of this work was to develop a biodegradable scaffold from a decellularized human umbilical cord (UC-scaffold) to heal full-thickness wounds. Decellularization was performed with 0.05% sodium dodecyl sulfate solution. The UC-scaffold was studied using morphological analysis methods. The composition of the UC-scaffold was studied using immunoblotting and Fourier transform infrared spectroscopy. The adhesion and proliferation of mesenchymal stromal cells were investigated using the LIVE/DEAD assay. The local reaction was determined by subcutaneous implantation in mice (*n* = 60). A model of a full-thickness skin wound in mice (*n* = 64) was used to assess the biological activity of the UC-scaffold. The proposed decellularization method showed its effectiveness in the umbilical cord, as it removed cells and retained a porous structure, type I and type IV collagen, TGF-β3, VEGF, and fibronectin in the ECM. The biodegradation of the UC-scaffold in the presence of collagenase, its stability during incubation in hyaluronidase solution, and its ability to swell by 1617 ± 120% were demonstrated. Subcutaneous scaffold implantation in mice showed gradual resorption of the product in vivo without the formation of a dense connective tissue capsule. Epithelialization of the wound occurred completely in contrast to the controls. All of these data suggest a potential for the use of the UC-scaffold.

## 1. Introduction

Wound healing is a complicated process comprising hemostasis, inflammatory reactions, cell proliferation, and tissue remodeling [1]. Deep and extensive non-healing skin wounds remain a serious medical problem that places a burden on patients, their families, and the healthcare system [2,3]. Autologous skin grafts for the treatment of full-thickness wounds are limited due to tissue availability, and other existing methods have only moderate clinical efficacy, justifying the need for the development of new approaches for the treatment of skin lesions [4,5].

Tissue engineering offers alternative ways to stimulate the healing of deep and extensive skin lesions. The biomimetics of the extracellular matrix (ECM) provide a microenvironment similar to that of the natural ECM, which is necessary for maintaining the vital activity and functional activity of cells at all stages of wound healing [6,7]. Previously, collagen, gelatin, hyaluronic acid, pectin, alginate, cellulose, chitosan, polyurethane, and silicone in the forms of films, fibers, hydrocolloids, hydrogels, sponges, and foams were used in technologies to restore damaged skin [8].

The effect of using extracellular matrix biomimetics is the formation of a microenvironment for cells that promotes the transmission of cellular signals and is involved in the regulation of local tissue homeostasis. Products made from decellularized tissues have the required properties and have shown satisfactory results upon their use [9,10,11,12,13]. The properties of decellularized tissues that promote the regulation of cellular functions and the modulation of tissue regeneration processes have been used in the development of tissue-engineered constructs. These tissue-engineered constructs are expected to be used to restore dysfunctional organs such as skin, cartilage, kidneys, liver, pancreas, lungs, and other tissues in experimental studies [7,13,14,15,16].

The preferred sources for obtaining scaffolds include the allogeneic tissues and organs of donors collected postmortem. However, the low availability of allogeneic material, due to ethical limitations and conflicts of interest with transplantation, does not allow their widespread use. In addition, the composition of allogeneic biomaterial is significantly affected by external and internal factors throughout the life of the donor, such as previous illnesses, developmental defects, and side effects of used medications [17,18].

The human umbilical cord as an extraembryonic tissue is devoid of the above disadvantages and, while preserving molecules of the fetal phenotype, contains a wide range of biologically active substances [19,20,21,22,23,24,25]:Different types of collagen, unsulfated and sulfated glycosaminoglycans (GAGs), and fibronectin;Numerous growth factors such as insulin-like growth factor (IGF) and protein binding insulin-like growth factors 1, 2, 3, 4, and 6; transforming growth factor alpha (TGF-α) and platelet growth factor (PDGF); fibroblast growth factors (FGFs); epidermal growth factors (EGFs); various isoforms of transforming growth factor beta (TGF-β1, 2, 3); vascular endothelial growth factor (VEGF); cytokines (with a predominance of anti-inflammatory cytokines); and matrix metalloproteinases and inhibitors of matrix metalloproteinases [23,24,26,27,28,29].

The expression of several immunomodulatory cytokines has also been detected, such as those that regulate activation, expression of normal T cells, and secretion (RANTES); interleukin-6 receptor; interleukin-16; and interferon gamma, as well as proinflammatory cytokines such as macrophage colonic stimulating factor, macrophage stimulating protein 1-alpha, superfamily receptors of tumor necrosis factor 1α and 1β, antagonist of interleukin-1 receptors, and cytokines associated with wound healing, including intercellular adhesion molecule 1 and granulocyte-stimulating factor [30]. The umbilical cord ECM has positive effects [20,31].

The decellularized umbilical cord stroma has been described as promoting the regeneration of cartilage [22], liver tissue [32], and nervous tissue [33]. In addition, part of the cell-engineered constructs with allogenic mesenchymal stromal cells ensures wound healing in rats [30,34]. The method of decellularization of biological tissues determines the composition, structure, and biological properties of the manufactured scaffold [9]. For example, the effect of the GAGs content in the scaffold of a decellularized cartilage on its functional activity as part of a cell-engineered construct has been shown [12].

The physical characteristics of scaffolds also have an impact on wound healing. In particular, sponges have high porosity and a high specific surface area, which accelerates tissue regeneration [23].

Experiments on in vivo animal models, despite the inability to fully reproduce the mechanism of human skin damage and the conditions of its healing, remain an indispensable stage in the research and development of new wound coatings [35]. Small laboratory animals remain the basis for exploratory research and verification of mechanisms that are crucial for the wound healing process [36]. To evaluate the functional properties of a biomimetic ECM based on a decellularized umbilical cord, it is advisable to use the in vivo model of a full-layer wound in mice with an anticontraction ring [37].

Therefore, the aim of the study was to develop a gentle and simple protocol for obtaining a biodegradable scaffold from a decellularized human umbilical cord (UC-scaffold) to heal full-thickness wounds.

## 2. Materials and Methods

### 2.1. Umbilical Cord Decellularization

Human umbilical cords (*n* = 5) were obtained from healthy full-term newborns after receiving informed consent from their mothers. The study was conducted in accordance with the guidelines of the Helsinki Declaration and approved by the Ethics Committee at the Kirov Military Medical Academy, St. Petersburg, Russia (17 December 2019, Protocol No. 230).

Under sterile conditions, after removing the vessels, the umbilical cords were crushed and homogenized (gentleMACS™ Dissociator, Milteniy Biotech, Bergisch Gladbach, Germany, program h-cord-01-01). The decellularization was carried out by treating fragments with 0.05% sodium dodecyl sulfate (SDS) solution (Biolot, Saint Petersburg, Russia) at RT using a shaker (Biosan, Rīga, Latvia) at 140 rpm for 24 h, followed by washing with a 500 mL phosphate–salt buffer (PBS) of pH 7.4 on mesh (Biolot, Saint Petersburg, Russia). Then, the UC-scaffold was lyophilized (VaCo 5-II, Zirbus, Bad Grund (Harz), Germany). Gamma irradiation at a dose of 1.5 Mrad was used to sterilize the UC-scaffold. The UC-scaffold was stored hermetically packed at a temperature of −20 °C. The flowchart for obtaining the UC-scaffold is presented in Figure 1.

### 2.2. DNA Quantification

The effectiveness of DNA removal in the UC-scaffold compared against native samples was evaluated by a DNA-DU-250 kit (Biolabmix, Novosibirk, Russia) in accordance with the manufacturer’s instructions. The native umbilical cord (*n* = 5) and decellularized tissues (*n* = 5) were dried on filter paper and weighed. The extracted DNA was then evaluated on a Nanodrop spectrophotometer (Thermo Fisher Scientific, Waltham, MA, USA) with an absorption coefficient of 260/280 nm. The DNA content was expressed as a ratio between the weight of DNA per tissue dry weight (ng/mg).

### 2.3. Biochemical Quantification of the GAGs Content

To measure the sulfated GAGs content, native tissue (*n* = 7) and the UC-scaffold (*n* = 7) samples were freeze-dried. The samples were digested overnight using papain solution (125 µg/mL papain in PBS at pH 6.5 with 5 mM L-cysteine hydrochloride and 5 mM ethylenediamine tetraacetic acid (all reagents were purchased from Sigma-Aldrich, St. Louis, MO, USA)). The digested samples were then centrifuged at 10,000× *g* for 10 min; 1,9-dimethylmethylene blue solution (200 µL) was added to each standard concentration of chondroitin sulphate and sample (20 µL) in a clear, flat-bottomed 96-well plate (all reagents were purchased from Sigma-Aldrich, St. Louis, MO, USA). The plate was agitated for 2 min before the absorbance was read on a Tecan infinite M200 Pro plate reader (Tecan Group, Männedorf, Switzerland) at a wavelength of 525 nm.

### 2.4. Electron Microscopy

The morphology of the surface and the nearest subsurface layer of samples were examined with scanning electron microscopy (SEM) using the JSM-7001F и JSM-6390 (JEOL, Tokyo, Japan) in the secondary electron mode at an accelerating voltage of 5 kV and a beam current of 10 pkA. To support charge leakage and enhance the secondary electron emission, a 30 nm thick gold layer was deposited by the Emitech K950 sputtering machine (Quorum Technologies, Lewes, UK).

The method of transmission electron microscopy (TEM) by Merlin (Carl Zeiss, Oberkochen, Germany) was used to study the microstructure of the samples. The samples of 15 mm in diameter and 2 mm in thickness were fixed in 2.5% glutaraldehyde for 12 h. Further, the preparations were carried out according to the standard procedure. The samples were immersed in 1% osmium tetroxide (Sigma-Aldrich, St. Louis, MO, USA) for 1 h, dehydrated with increasing ethanol concentration, and dried. Then, they were embedded into epoxy (Araldite, Cambridge, UK).

### 2.5. Infrared Spectroscopy

Spectra were recorded in reflection mode between 4000 and 500 cm^−1^ on a Bruker Vertex (Bruker Optics, Ettlingen, Germany) for 45 scans conducted on different areas of the lyophilized samples. The resolution of a single beam spectra was 2 cm^−1^. Measurements were performed at +21–± 1 °C in an air-conditioned room.

The infrared spectra of the native umbilical cord and the UC-scaffold were recorded using an IRPrestige-21 Fourier-transform infrared spectrometer (FTIR) (Shimadzu, Kyoto, Japan) in transmission mode and averaging 100 scans in the range of 4500–650 cm^−1^, with a spectral resolution of 2 cm^−1^.

### 2.6. Western Blotting Analysis

The samples dissolved in 6 M urea were diluted in a 1:1 ratio in a Laemmli buffer (10 mL of glycerol; 4 mL of 1.5 N Tris-HCl, pH 6.8; 2 g of SDS; 0.002 g of bromophenol blue; and 5 mL of β-mercaptoethanol (all reagents DIA-M, Moscow, Russia)) and incubated at +90 °C for 5 min.

The proteins in the obtained samples were separated by polyacrylamide gel electrophoresis in the presence of SDS on the Bio-Rad system (Bio-Rad Laboratories, Hercules, Contra Costa County, CA, USA) and transferred to polyvinylidene fluoride membranes (Sigma-Aldrich, St. Louis, MO, USA) in a Tris–glycine buffer (pH 8.3) containing 10% ethanol using wet transfer systems (Bio-Rad Laboratories, Hercules, Contra Costa County, CA, USA). After transfer, the membranes were blocked for 1 h at RT with a 5% solution of bovine serum albumin in a PBS containing 0.05% Tween 20 (PBS-T) and then incubated overnight at +4 °C in the presence of one of the following primary antibodies: mouse monoclonal antibodies against TGF-β3 (MAB949 Hu22, Cloud-Clone Corp., Tsingtao, USA, 1:2000), mouse monoclonal antibodies against VEGFA (#MHD12601, Affinity Biosciences, Taiwan, China, 1:2000), or rabbit polyclonal antibodies against type I collagen (AF0134, Affinity Biosciences, China, 1:200). After repeated washing with PBS-T, the membranes were incubated in the presence of goat antibodies conjugated with horseradish peroxidase against mouse or rabbit immunoglobulins (1:10,000 in PSB-T, Pierce, Waltham, MA, USA), for 1 h at RT. Then, the membranes were washed with PBS-T and detected using a SuperSignal™ substrate West Dura Extended Duration Substrate (Thermo Fisher Scientific). Chemiluminescence was recorded using the ChemiDoc Imaging System (Bio-Rad Laboratories, Hercules, Contra Costa County, CA, USA).

### 2.7. Determination of Swelling Degree of the UC-Scaffold

To study the ability of the UC-scaffold to absorb moisture, it was lyophilized, micronized, and sifted through a 1 mm sieve. An amount of 100 mg of the UC-scaffold (*n* = 10) was immersed in 10 mL of 0.9% NaCl solution for 1 h at +37 °C and weighed at predetermined time intervals, with the UC-scaffold weight and volume increasing due to the swelling process. The equilibrium swelling ratio (%) of the UC-scaffold was calculated based on the equation below:Swelling ratio (%) = (Ws − Wd)/Wd × 100,(1)
where Ws is the weight of the swollen UC-scaffold, and Wd is the dry weight of the UC-scaffold.

### 2.8. In Vitro Scaffold Biodegradation Assay

The study of the effect of 0.1% collagenase solutions of 159 U/mL (Biolot, Saint Petersburg, Russia) and hyaluronidase of 1280 U/mL (Microgen, Moscow, Russia) on the UC-scaffold was carried out with the registration of the results on weight loss of samples, and the spectral characteristics of the FTIR were investigated.

The UC-scaffold samples (*n* = 9) were weighed in microsamples (Eppendorf, Hamburg-Nord, Germany). Collagenase or hyaluronidase solutions were added to the samples, and PBS of pH 7.4 was added to the control samples in a ratio of 1:20. Enzyme lyophilizates were dissolved with sterile PBS and incubated at +37 °C with closed lids. At 24 and 72 h after the start of incubation, the samples were cooled, and the supernatant was extracted (LMC-4200R BioSan, Rīga, Latvia; 1500 rpm). The precipitate was lyophilized for 48 h (ZirbusVaCo5II, Bad Grund (Harz), Germany) and weighed. The weight loss of the product was expressed as a percentage of the initial mass of the sample. Next, the lyophilized samples were examined by the FTIR method.

### 2.9. hADSCs Isolation and Cell Culture Maintenance

Adipose tissue samples weighing 3–5 g were obtained with the informed patient consent of one living healthy donor during liver transplantation under general anesthesia. The study was conducted in accordance with the guidelines of the Helsinki Declaration and approved by the Local Ethics Committee at the Shumakov National Medical Research Center of Transplantology and Artificial Organs, Moscow, Russia (15 November 2019, Protocol No. 151119-1/1e). The tissue was incubated in 0.1% collagenase solution type I at +37 °C for 20 min. The human adipose-derived mesenchymal stem cells (hADSCs) were cultured in growth medium (DMEM/F12 (1:1) with the addition of 10% fetal cattle serum, 100 of U/mLpenicillin, 100 μg/mL of streptomycin sulfate, and 2 mMof L-glutamine (all reagents, Gibco™, Thermo Fisher Scientific). In the experiment, third-passage cells were used.

The immunophenotypic expression profile of cell markers isolated from the adipose tissue met the criteria of The International Society for Cellular Therapy and confirmed that these cells were multipotent hADSCs [37]. In earlier studies, we found high levels of expression of CD29, CD44, CD49b, CD73, and CD90 in the primary culture. At the same time, the expression of CD34, CD45, or HLA-DR was not observed in the culture [38].

### 2.10. Viability of hADSCs Cultured on the UC-Scaffold

Samples consisted of 5 × 10^5^ hADSCs and 10 mg of the UC-scaffold. The UC-scaffolds, pre-soaked during the 24 h in the culture medium, were seeded with cells in tubes using the Multi Bio 3D orbital shaker (Biosan, Riga, Latvia). The cell viability of the hADSCs cultured on the UC-scaffold was assessed after 7 and 14 days with the standard protocol of LIVE/DEAD™ assay (Invitrogen™, Thermo Fisher Scientific). The solution consisted of 1 µM of calcein AM and 4 μM of ethidium homodimer prepared in PBS. Briefly, hADSCs cultured on the UC-scaffold were soaked in a staining solution for 30 min at +37 °C. The samples were washed twice with PBS before imaging by Nikon Ti (Nikon, Tokyo, Japan) using λex = 490/λem = 515 nm for calcein AM and λex = 495/λem = 635 nm for ethidium homodimer.

### 2.11. In Vivo Study

The manipulations did not cause pain to the animals and were carried out in compliance with the following Russian legislation [39,40,41]. The Ethics Committee at the Kirov Military Medical Academy (Saint Petersburg, Russia) approved this study (31 May 2022, Protocol No 263). Anesthesia was induced by the intramuscular administration of 15 mg/kg of Zoletil 100 (Virbac, Carros, France). After surgery, no antibiotic treatment was administered. Three-month-old mongrel white mice (both sexes, with an equal number of males and females) weighing 24.25 ± 0.48 g were used in the experiments. The duration of quarantine (acclimatization period) was 14 days for all the animals.

The UC-scaffold was subcutaneously implanted in the mongrel mice (*n* = 20). The animals (*n* = 20) were aseptically cut to a length of 5 mm at the withers, and 0.01 g of the sterile UC-scaffold was implanted under the skin. The incisions were sewn with 6-0 Surgipro II surgical thread (Medtronic, Minneapolis, MN, USA). On the 3rd, 7th, 14th, and 21st days of the experiment, five animals were removed for further histological analysis.

The study of the effect of the UC-scaffold on the dynamics of healing of full-layered skin was carried out on mongrel mice (*n* = 64). Under aseptic conditions, a full-layer skin wound with a diameter of 1.2 cm was applied to mice with a scalpel on the withers area, the fur of which had been removed. Silicone rings with a diameter of 1.4 cm were sewn to the edges of skin defects of randomly selected animals of the experimental group, and a UC-scaffold (0.01 g) soaked with 0.2 mL of sterile PBS solution was placed in the wound. To prevent drying and contamination, the defect was closed in a contactless manner with a paraffin film. The animals of the control group (*n* = 32) underwent similar manipulations but without placing a UC-scaffold in the wound. The duration of animal follow-up after the sample administration was 1, 3, 7, 14, and 21 days.

The samples were preserved in a 10% formalin solution, dehydrated, paraffinized, sectioned at 4–5 μm, deparaffinized in xylene, rehydrated, and stained with hematoxylin and eosin (H&E) and with Heidenhain’s staining (Biovitrum, Saint Petersburg, Russia).

### 2.12. Histological and Immunohistochemical Staining Staining

Samples of a lyophilized UC-scaffold and a native umbilical cord were preserved in a 10% formalin solution, dehydrated, paraffinized (station for filling biological tissues with HistoStar paraffin, Thermo Fisher Scientific), sectioned at 4–5 µm (automatic rotation microtome complete with STS HM 355S, Thermo Fisher Scientific), deparaffinized in xylene, dehydrated, and stained with H&E and Alcian blue (pH 2.5) (Biovitrum, Saint Petersburg, Russia). To control the staining of GAGs, individual sections were incubated in a 1280 U/mLhyaluronic acid solution (Microgen, Moscow, Russia) for 24 h at +37 °C, followed by staining with Alcian blue. After dehydration, the preparations were covered with cover glasses (Thermo Fisher Scientific). The staining results were evaluated at different magnifications using a Carl Zeiss AxioScope.A1 microscope (Carl Zeiss, Oberkochen, Germany).

For immunohistochemical examination for the presence of type IV collagen and laminin in the UC-scaffold, the dewaxed sections were exposed to antigen demasking in 10 mM tris-EDTA buffer at pH 9.0. Then, the specific antigen expression was determined using the EnVision peroxide-based system (Dako, Glostrup, Denmark) according to the manufacturer’s instructions. Non-specific binding sites were blocked using 10% normal goat serum (Dako, Glostrup, Denmark). The tissues were incubated for 2 h with primary mouse antibodies directed against human laminin (1/50; LAM-89, Leica, Berlin, Germany) and type IV collagen (1/25; CIV22, DAKO, Glostrup, Denmark). To visualize the binding of antibodies, a substrate of the diaminobenzidine chromogen was added, after which the tissues were contrasted with Mayer hematoxylin (Biovitrum, Saint Petersburg, Russia).

Polyclonal rabbit antibodies (1/50; AF0738; Affinity Biosciences, China) in an antibody solution (Diagnostic BioSystems, Pleasanton, CA, USA) were used to identify fibronectin in the UC-scaffold. The antigen was unmasked in a microwave oven with a 10 mM Tris-EDTA buffer (pH 9.0; Diagnostic BioSystems) for 15 min, followed by cooling at room temperature, washing with PBS, and blocking non-specific binding (Diagnostic BioSystems) according to the manufacturer’s instructions. Polyclonal anti-rabbit antibodies conjugated with Alexa Fluor 488 (1/1000; SAA544Rb11; Cloud-Clone Corp., Tsingtao, China) were used for visualization.

Staining without primary antibodies served as a negative control. Then, the stained sections were dehydrated and covered with cover glasses. To fix the staining results, light microscopy (Axio Scope.A1, Carl Zeiss, Oberkochen, Germany) employing diaminobenzene chromogen and fluorescence microscopy (Axio Observer, Carl Zeiss, Oberkochen, Germany) were carried out using Alexa Fluor 488.

### 2.13. Statistical Analysis

The data were analyzed with the SPSS 26.0 statistical software package. The results were presented as mean ± SD. The distribution of the variables was tested with the Shapiro–Wilk procedure. The results were compared using *t*-testing, one-way analysis of variance (ANOVA), and Tukey’s honestly significant difference post hoctest or Bonferroni posthoc test, where *p* < 0.05 was considered statistically significant.

## 3. Results and Discussion

Perinatal tissues are an attractive substrate for the development of bioactive scaffolds employing them as a base and intended for use in the clinic due to their multicomponent composition and availability; they are disposed of after childbirth as biological waste [25]. The objectives of this study included the characterization of the composition and properties of the UC-scaffold we obtained.

### 3.1. The UC-Scaffold Characterization

Published research data demonstrate that the UC-scaffold is biocompatible and has a positive effect on regeneration processes in different tissues [20,21,22,23,24,26,27,28,29,30,32,33,42,43,44,45,46,47]. Different research groups use different protocols for removing cells, DNA, and cellular debris from the human umbilical cord [20,21,22,23,24,26,28,29,30,32,33,43,44,45,46,47,48].

Researchers have used a combination of physical, chemical, and enzymatic methods to decellularize Wharton’s jelly [20,21,22,23,24,26,28,29,30,32,33,43,44,45,46,47]. D. Li et al., S. Jadalannagari et al., G. Converse et al., L. Penolazzi et al., Z. Yuan et al., and M. Dubus et al. obtained a UC-scaffold using surfactants (a unique combined treatment protocol in each research group) and then applied nuclease treatment [20,21,26,43,44,45,46]. N. Azarbarz et al. and F. Ramzan et al., on the contrary, after the application of enzymatic treatment with trypsin solution, used surfactant treatment [22,28]. Z. Koci et al. and K. Vyborny et al. also treated Wharton’s jelly with trypsin, which was followed by incubation in acetic acid and ethanol [33,47]. F. Ramzan et al., Z. Koci et al., and K. Vyborny et al. produced hydrogel from the obtained UC-scaffold by treatment with pepsin [22,33,47]. K. Foltz et al., J.-H. Lu et al., M. Kehtari, and B. Beiki et al. applied a more sparing regimen of chemical exposure to Wharton’s jelly and used SDS solutions (the concentration in each study is dissimilar) [29,30,32,33]. Furthermore, M. Kehtari et al. and B. Beiki et al., after decellularization, used the procedure of chemical cross-linking of the resulting product [23,32]. When creating a scaffold for tissue engineering of vessels from silk fibroin, P. Gupta et al. functionalized it with water-soluble components of the UC-scaffold [24]. A. Basiri et al. added a water-soluble product obtained by a similar technology to a silk fibroin hydrogel before gelation [27].

The technology used in our study to remove the cells of the Wharton’s jelly was chosen, first, in order to avoid the use of enzymatic decellularization agents, which may remain in the UC-scaffold and resume their activity after its implantation. Second, to minimize the effect of surfactants on the scaffold, they were used in a minimum concentration, and thorough washing with a PBS solution was carried out, thereby maximally preserving the natural composition of the tissue, as well as increasing biocompatibility and reducing the risk of maintaining excessive amounts of surfactants in the decellularized tissue [49].

On histological samples of the native umbilical cord stained with H&E (Figure 2a), a large number of cell nuclei were visualized, while on the UC-scaffold samples, there were no cell nuclei (Figure 2b). The residual DNA content in the UC-scaffold was 21.8 ± 5.0 ng/mg of dry weight, which was statistically significantly lower (*p* = 0.000001) than the DNA content in the native umbilical cord of 506.8 ± 39.1 ng/mg of dry tissue (Figure 2c) and met the minimum criteria for the residual content of the donor’s genetic material [50]. Thus, the developed technology of decellularization of the human umbilical cord stroma ensured the removal of about 96% of DNA, which indicated its effectiveness and, accordingly, are duced likelihood of the scaffold causing undesirable immune reactions in the recipient.

The topography of the implanted UC-scaffold surface and its microstructure can influence the functional activity of recipient cells [38,51]. Porosity and surface roughness are important parameters for cell migration, adhesion, and proliferation on the scaffold surface. The rough porous surface of the UC-scaffold (Figure 2d,e) has been demonstrated. The heteroporosity of the UC-scaffold contributes to the circulation of liquids and nutrients, as well as to the colonization of the scaffold by recipient cells [52].

The bulk of the ECM of the umbilical cord connective tissue is made up of collagens, which is demonstrated on micrographs of TEM native tissue (Figure 2f,g). After decellularization of the umbilical cord, the UC-scaffold retains collagen fibers in its composition. This fact is shown in micrographs of TEM (Figure 2h,i) and is confirmed by FTIR spectra close to those described in the literature for collagen spectra (Table 1) [22,53,54,55].

Strong signals in the Amide I region indicate the presence of carboxyl groups. A strong signal in the Amide II region corresponds to the valence and deformation vibrations of amino groups (Table 1) [21]. The immunoblotting method showed that type I collagen formed the main part of the structural collagens of the native umbilical cord and the UC-scaffold (Figure 2j).

One of the functions of Wharton’s jelly is to ensure blood flow through the umbilical cord vessels, even under conditions of twisting and squeezing during intrauterine fetal development [25]. This is possible due to the high content of GAGs embedded in the collagen network [25,56]. The preservation of GAGs after decellularization was confirmed by histochemical staining with Alcian blue (Figure 2k,l) and confirmed by the absence of such staining after treatment of the preparations with hyaluronidase solution (Figure 2m). The spectral characteristics of FTIR in the absorption regions between 1170 and 800 cm^−1^ not only are responsible for carbohydrate fragments of collagen but also indicate the presence of the UC-scaffold GAGs (Table 1) [21].

The effectiveness of decellularization is determined by the completeness of DNA removal and the preservation of not only the microstructure of collagen but also sulfated GAGs. Negatively charged molecules of sulfated GAGs, interacting with growth factors, contribute to their slow release and constant local concentration at the site of scaffold implantation. This has a special significance for cell activation and stimulation of healing [53]. It was found that the UC-scaffold contained 45.72 ± 2.44 µg/mg dry weight of sulfated GAGs, while in the native umbilical cord their number was 36.98 ± 0.59 µg/mg dry weight (*p* = 0.05) (Figure 2n). This fact was explained by an increase in the proportion of GAGs per unit mass of dried tissue due to the removal of umbilical cord cells, which was consistent with previously published data [20,22].

The presence of type IV collagen and laminin (Figure 2o,p) was shown in the composition of the UC-scaffold by immunohistochemical staining (Figure 2o,p). Being a complex adhesive protein, laminin, together with non-fibrillar type IV collagen, participates in the formation of a network of the skin’s basement membrane [57,58].

Fibronectin (Figure 2q) found in the UC-scaffold is of particular importance for healing skin wounds. This glycoprotein exposes the RGD sequence of amino acids, which binds to the α_V_β_3_-integrin of fibroblasts, thereby stimulating their migration into the wound. Fibronectin has also been shown to regulate the proteolytic activity of lysyl oxidase and collagen formation [57,59].

A unique feature of fetal phenotype tissues, which also include the umbilical cord, is the high content of the antifibrotic isoform TGF-β3 in them, relative to the isoforms TGF-β1 and TGF-β2 inherent in postnatal tissues [60]. The presence of TGF-β3 in the UC-scaffold was also confirmed by the immunohistochemical method (Figure 2r). Vascular endothelial growth factor (VEGF) was also found in the UC-scaffold (Figure 2s). The presence of these growth factors is important for stimulating the formation of functionally active tissue in the lesion area [61,62].

Implantation into the body of a scaffold is accompanied by the migration of recipient cells to it by their proliferation, followed by replacement of damage with functionally active tissue [34]. To simulate the process of colonization by the UC-scaffold cells, we used the most readily available cell population of human origin—hADSCs. LIVE/DEAD fluorescent staining made it possible to determine a significant mass of viable cells on the surface of the scaffold, while the proportion of dead cells in the sample was comparatively little (Figure 2t,u). By the third day of cultivation, most of the cells were successfully attached to the surface of the scaffold and proliferated (Figure 2t). On day 14, we observed a significant increase in cell mass. Most of the cells had a fibroblast-like morphology characteristic of the normal development of hADSCs’ cell culture (Figure 2u);the UC-scaffold did not have a cytotoxic effect on human dermal fibroblasts and the cells of various animal organs [63]. Thus, the absence of cytotoxicity of the UC-scaffold was proved, and its ability to maintain cell adhesion was confirmed. This indicates the prospects for further research in the field of tissue engineering using the UC-scaffold.

The swelling ratio was a quantified datum that was regarded as the capacity of the UC-scaffold to absorb moisture. The ability of theUC-scaffold to absorb moisture is an important feature of scaffolds used in tissue engineering and directly related to the degree of porosity of the scaffold. In the first 10 min, all samples of the UC-scaffold exhibited a rapid moisture absorption capacity. The swelling ratio of the UC-scaffold was 1617 ± 120%. The scaffold loaded into the solution swells until the osmotic forces contributing to the stretching of the collagen polymer mesh are balanced by the elastic forces of the stretched polymer segments [22]. The porous rough morphology of the scaffold and the presence of sulfated forms of GAGs in it contribute to moisture absorption, provide sufficient area for cell growth and migration, and increase the absorption of exudate formed in the wound [21].

Fluid absorption is an important criterion for wound coatings. Plasma exudation into the wound is a natural and obligatory stage of the inflammatory process. The formation of exudate in the wound occurs because of the release of the liquid part of the blood into the tissue due to venous stagnation and ensures the migration of effector cells to the inflammation site and self-cleaning of the wound. However, wounds from which a large amount of fluid is released require special attention. With excessive exudate, wound healing can be complicated by suppuration and deterioration of microcirculation. Therefore, the use of exudate adsorbing and air-permeable agents in the early stages, which do not inhibit the pathophysiological processes of self-purification, is favorable for wound healing [64,65].

The biodegradation ability of the UC-scaffold was shown in vitro (Figure 3a,b). The natural biodegradation of the implanted UC-scaffold eventually eliminates the need to form a tissue response to a “foreign body” (well described in the context of synthetic implants) or its encapsulation [66]. For many applications in regenerative medicine, it will be necessary to prolong the biodegradation time; for this, it is advisable to try out various innovations for the UC-scaffold lysate cross-linking.

Incubation with collagenase for 72 h resulted in a 94.3 ± 0.9% weight loss of the samples (Figure 3c). The FTIR spectra of lyophilized scaffold samples subjected to collagenase cleavage demonstrated differences (Figure 3a). The main peaks are also shown in the Amide I and Amide II regions (Table 1). There was a shift of Amide A and B to the right side of the spectrum and of Amide I, II, and III to the left. At the same time, after three days of incubation in collagenase solution, there was no distinct peak in the 1203–1205 cm^−1^ spectral region. The peaks responsible for carbohydrate fragments of collagen and GAGs persisted after 72 h of incubation with collagenase but were slightly shifted to the left.

After 72 h of incubation of the scaffold with hyaluronidase solution, there was no significant weight loss of the samples (Figure 3c). The FTIR spectra of lyophilized scaffold and scaffold samples after 72 h of incubation with hyaluronidase demonstrated peak differences in spectral regions, the severity of which was affected by the presence of GAGs. After exposure to hyaluronidase, a decrease in absorption was observed in the spectral range 1135–700 cm^−1^. Absorption was particularly noticeable in the peak regions of 1078 cm^−1^, 1032 cm^−1^, and 885 cm^−1^, which confirmed the absence of complex carbohydrates (Figure 3b). The absorption intensity in the spectral region of Amide II clearly increased. A slight shift of peaks attributed to the Amide A and B regions was noted. Amide I and Amide II also shifted to the right. Peak 1397 cm^−1^ was isolated after incubation in hyaluronidase. Peaks attributed to Amide III were detected in the same spectral regions.

The obtained results indicated that the composition of the UC-scaffold included collagens (mainly type I collagen), GAGs, components of the basement membranes (type IV collagen and laminin), fibronectin, and growth factors (VEGF and TGF-β3); was biodegradable; and had the property of absorbing moisture (and, hence, wound exudate). In addition, proliferation of hADSCs on the UC-scaffold was shown. These properties of the scaffold are necessary for the biocompatibility of the UC-scaffold, designed to stimulate skin regeneration and its integration into the surrounding tissues of the recipient.

### 3.2. In Vivo Biocompatibility of the UC-Scaffold

Subcutaneous implantation of the UC-scaffold in mice showed gradual resorption of the product. After the operation, all animals showed no signs of depression throughout the observation period. The healing of the skin incisions created for the implantation of the UC-scaffold fragment occurred without complications or visible signs of inflammation and edema. Xenografts (i.e., human umbilical cord) were well tolerated by immunocompetent mice. Histomorphological examination of the explanted UC-scaffold did not reveal the formation of a single tissue capsule or the presence of multiple giant cells of “foreign bodies” in the tissues of the implantation area (Figure 4).

In preparations stained with H&E, the formation of granulation tissue typical of the normal wound healing reaction around the UC-scaffold explant (Figure 4a,c,e,g) was visible [63]. Heidenhain trichrome staining revealed the presence of collagen fibers in the area of the UC-scaffold implantation on day 14, which indicated its gradual resorption (Figure 4b,d,f,h).

Based on the data obtained in the experiment of subcutaneous implantation of the UC-scaffold in vivo, it can be concluded that the UC-scaffold is biocompatible and well tolerated by the recipient’s body.

### 3.3. Wound Healing Activity of the Fabricated UC-Scaffold

The use of cadaveric biomaterial has a number of disadvantages for the treatment of full-thickness skin and soft tissue injuries. The total effects of the action of external factors, the intensity of metabolic processes, and the functioning of the body’s regulating systems affect the skin during the life of the donor, as well as the composition of ECM and the ratio of its components. A decrease in the content of type IV and VII collagens contributes to the weakening of the connections between the dermis and the epidermis. The addition of glucose and fructose molecules to collagens and ECM proteins inhibits their contractility and makes them resistant to remodeling. The processes of photoaging and oxidative stress are the cause of collagen fragmentation and chemical modification of amino acid residues. A decrease in the fibronectin content worsens the binding of cells through integrin receptors to ECM [67]. Age-related changes in donor tissues inevitably worsen the properties of the scaffold made from them and reduce their regenerative potential.

Since the maximum age of large animal donors of biological material is significantly less than the age of a human (acting as a donor of biomaterial to create a decellularized product based on it), many researchers use xenogenic tissues. Internationally, research is being carried outon the creation of scaffolds from organs and tissues of animal origin [10,68]. However, there are risks of immune reactions (due to interspecific protein differences) [69]. Interest in the use of extraembryonic tissues (including the placenta, amniotic membrane, and umbilical cord) as raw materials for use in regenerative medicine has resumed since the 1990s [25,70]. To produce a scaffold with optimal properties that stimulate regeneration, we chose the biomaterial of the human umbilical cord.

To study the effect of biomaterials acting on the entire set of complex pathophysiological processes occurring during wound healing, an appropriate alternative choice is the use of in vivo models. Traditionally, small rodents are used to model full-thickness skin wounds [71]. In vivo experiments were conducted on a model of a full-thickness skin defect in experimental mice (Figure 5). The aim of these experiments was to evaluate the effectiveness of the lyophilized form of the UC-scaffold. Because the skin of rodents has distinct features—namely, wound healing in rodents occurs mainly due to the contraction of the panniculus carnosus layer—additional devices were used to limit the contraction of wound edges [37,71].

During the first day after the primary alteration, minor differences between the experimental and control groups were observed regarding the vascular reaction in the tissues surrounding the defect area (Figure 5a–d). The expansion of the lumen of arterioles and capillaries with the phenomena of stasis and perivascular edema with the emigration of granulocytes in the defect area was observed. The UC-scaffold was visible in the wounds of mice of the experimental group.

On the third day after surgery, the formation of granulation tissue was observed in both the experimental and control groups (Figure 5e–h). The UC-scaffold was embedded in the scab covering the wound (Figure 5e,f). At this stage of the study, moderate infiltration of mononuclear phagocytes in the tissues of the two groups was observed.

As a typical pathological process, an inflammatory reaction develops in tissues with a developed vascular network in response to the action of a phlogogenic agent and manifests itself in successive phenomena aimed at eliminating the etiological factor and restoring damaged tissue. In accordance with the principle of autochthonicity, once started, inflammation proceeds through all stages regardless of the action of the damaging agent [72]. Short-term vascular spasm at the time of injury is replaced by the phenomena of arterial and then venous hyperemia. Disorganization of vascular wall components by lysosomal enzymes of damaged cells and metallomatrix proteinases increases vascular permeability for the initial emigration of polymorphonuclear leukocytes and macrophages. The release of inflammatory mediators and the expression of adhesion molecules cause the “marginal standing” of neutrophilic leukocytes, their migration through the vessel wall, and the accumulation of chemoattractants and chemokines—their directional movement outside the vessel, which is facilitated by exudate. During the first day, the processes of slowing down blood flow ensure localization of the injury area and self-cleaning of the wound by phagocytosis. The bactericidal action of phagocytes in the wound is carried out mainly by means of reactive oxygen species. However, their excessive activity can have a negative impact on the healing process [72].

By the seventh day after implantation of the UC-scaffold in the animals of the experimental group, there was a more intensive filling of the wound defect space with granulation tissue (Figure 5i,j), compared with the control group, in mice with an anti-contraction ring but without the UC-scaffold in the wound (Figure 5k,l). Fragments of the UC-scaffold were visible in granulation tissue and were not surrounded by phagocyte cells.

Two weeks later, an epithelial layer was formed over the wound defects containing the UC-scaffold (Figure 5m,n), unlike the wounds in the control group (Figure 5o,p). We did not detect any intense inflammatory reactions in the wound areas nor cellular reactions to the presence of a “foreign body” in the animals of the experimental group. The gradual degradation of the UC-scaffold placed in wounds was accompanied by the active filling of the wound defect bed with granulation tissue and its maturation. On the 14th day of the experiment, the areas of tissue defects in which the UC-scaffold was placed were completely filled with granulation tissue. In the control group, the wound area was also filled with granulation tissue, but its height was not uniform.

The lyophilized UC-scaffold placed in the wound absorbed the released exudate and partially filled the tissue defect, replacing the ECM missing in the wound and gradually integrating into the surrounding tissues. At the same time, the UC-scaffold did not significantly affect the course of the natural processes of exudation and phagocytosis, which was manifested in the absence of excessive (relative to control group) accumulation of neutrophils near the UC-scaffold.

The results of an experiment to study the functional activity of the UC-scaffold in vivo showed that the healing of full-thickness skin wounds in mice in the presence of the UC-scaffold was characterized by moderately pronounced processes of alteration, vascular reactions, and selective focusing of leukocytes close to the control group. At the same time, the presence of the UC-scaffold in the wound was already accompanied by active formation of granulation tissue on the seventh day (Figure 5i,j). On the 14th day, filling of the wound containing the UC-scaffold with granulation tissue and epithelialization were revealed in contrast to the control group (Figure 5m–p).

The obtained results of in vitro and in vivo experiments allow us to assert the regenerative potential of the resulting UC-scaffold.

Various forms of medical products, such as films, fibers, sponges, hydrogels, and fine particles, can be manufactured on the basis of the UC-scaffold in the future. The crushed powdered forms can be conjugated with pharmacological agents to achieve more complex drug release profiles. Mixing powdered products from the UC-scaffold with inorganic materials that provide the product rigidity can be tested in the field of filling bone defects. In some cases, the use of injectable forms of ECM hydrogels, which are capable of in situ polymerization, is more appropriate since such materials can easily adapt to the surface relief and volume of the damaged area [22]. A hydrogel based on the UC-scaffold can be used for 3D-printing technology [14].

## 4. Conclusions

As a result of the conducted research, the possibility of obtaining a scaffold from the human umbilical cord, which promotes the healing of full-thickness skin lesions, has been shown. The lyophilized form of the scaffold is preferable due to its high ability to adsorb wound exudate, its preservation of biological activity, and the possibility of sterilizing the product in dry form. Other important advantages of the UC-scaffold lyophilizate are the possibility of long-term storage under normal conditions and it being ready-made. The manufacturing technology of the UC-scaffold is relatively simple andinexpensive and allows for the creation of a homologous product. The chosen mode of umbilical cord decellularization allowed, with the removal of 96% of DNA, the preservation of sufficient amounts of GAGs (including sulfated GAGs), components of the basement membranes (type IV collagen and laminin), cell adhesion molecules, and growth factors (fibronectin, VEGF, and TGF-β3). The porosity of the UC-scaffold ultrastructure provided the possibility of its settlement by the cells of experimental animals.

The UC-scaffold is able to completely degrade, which has been shown in in vitro experiments, since its bulk consists of collagens and GAGs. Preserving the native fiber structure of the human umbilical cord stroma after decellularization makes it possible to create the most biocompatible product.

The testing of the effect on the healing processes of full-layered skin wounds in animals showed biocompatibility, biointegration, and the regenerative potential of the UC-scaffold.

Thus, the developed UC-scaffold has high regenerative potential, is able to create the necessary microenvironment for cells, and can be considered a promising material for creating a wound healing product.

## Figures and Tables

**Figure 1 biomimetics-09-00405-f001:**
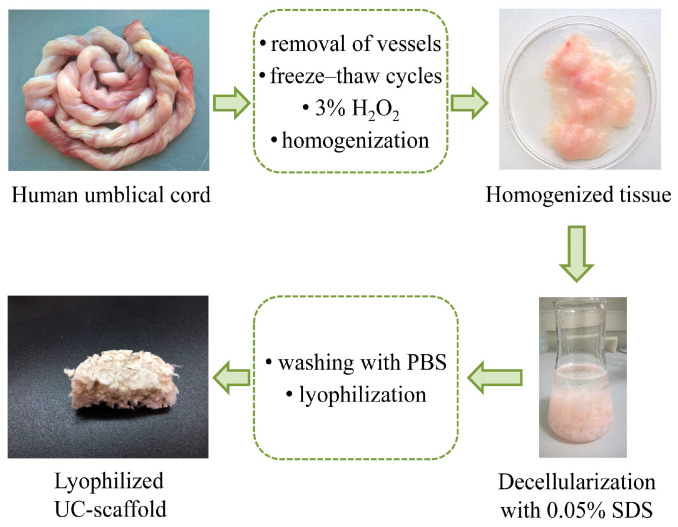
A flowchart of a decellularization procedure for human umbilical cords.

**Figure 2 biomimetics-09-00405-f002:**
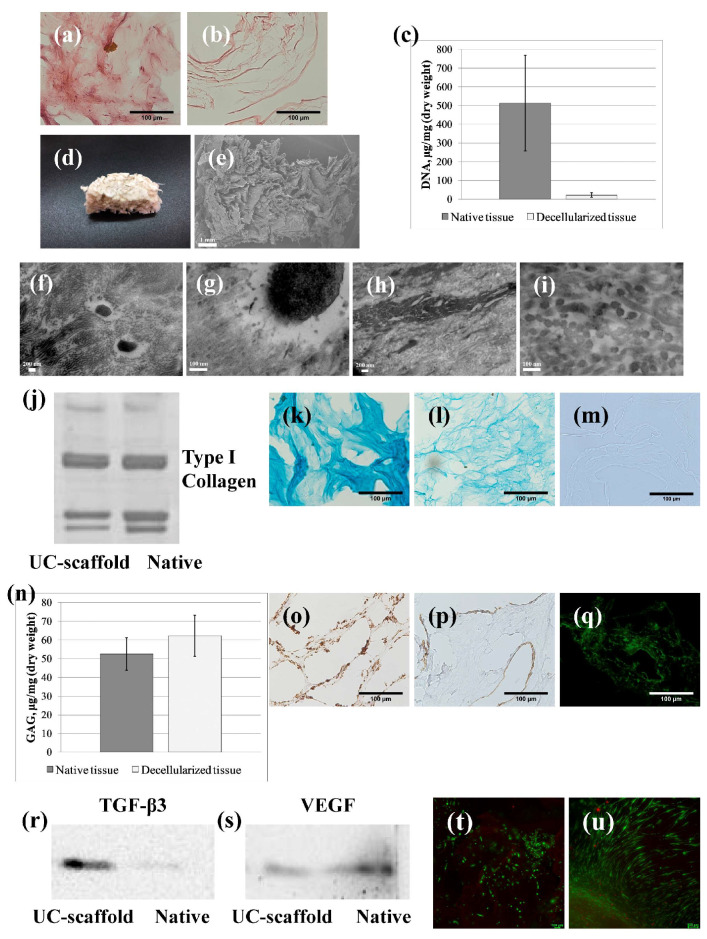
A characterization of the native and decellularized human umbilical cord (UC-scaffold): (**a**) native tissue H&E staining (scale bar = 100 µm); (**b**) the UC-scaffold H&E staining (scale bar = 100 µm); (**c**) DNA content, ng per 1 mg of dry weight; (**d**) the macroscopic appearance of the scaffold; (**e**) scanning electron microscopy (SEM) image of the scaffold (scale bar = 1 mm); (**f**) transmission electronic microscopy (TEM) image of the native tissue (scale bar = 200 nm); (**g**) TEM image of the native tissue (scale bar = 100 nm); (**h**) TEM image of the scaffold (scale bar = 200 nm); (**i**) TEM image of the native tissue (scale bar = 100 nm); (**j**) content of type I collagen as measured via Western blotting; (**k**) native tissue Alcian blue staining (scale bar = 100 µm); (**l**) the UC-scaffold Alcian blue staining (scale bar = 100 µm); (**m**) Alcian blue staining after the UC-scaffold treatment with hyaluronidase (scale bar = 100 µm); (**n**) GAG content, µg per 1 mg of dry weight; (**o**) immunohistochemical staining for type IV collagen (scale bar = 100 µm); (**p**) immunohistochemical staining for laminin (scale bar = 100 µm); (**q**) immunohistochemical staining for fibronectin (scale bar = 100 µm); (**r**) content of TGF-β3 as measured via Western blotting; (**s**) content of VEGF as measured via Western blotting; (**t**) scaffold with hADSCs, 7 days of cultivation, LIVE/DEAD staining (scale bar = 100 µm); and (**u**) scaffold with hADSCs, 14 days of cultivation, LIVE/DEAD staining (scale bar = 100 µm).

**Figure 3 biomimetics-09-00405-f003:**
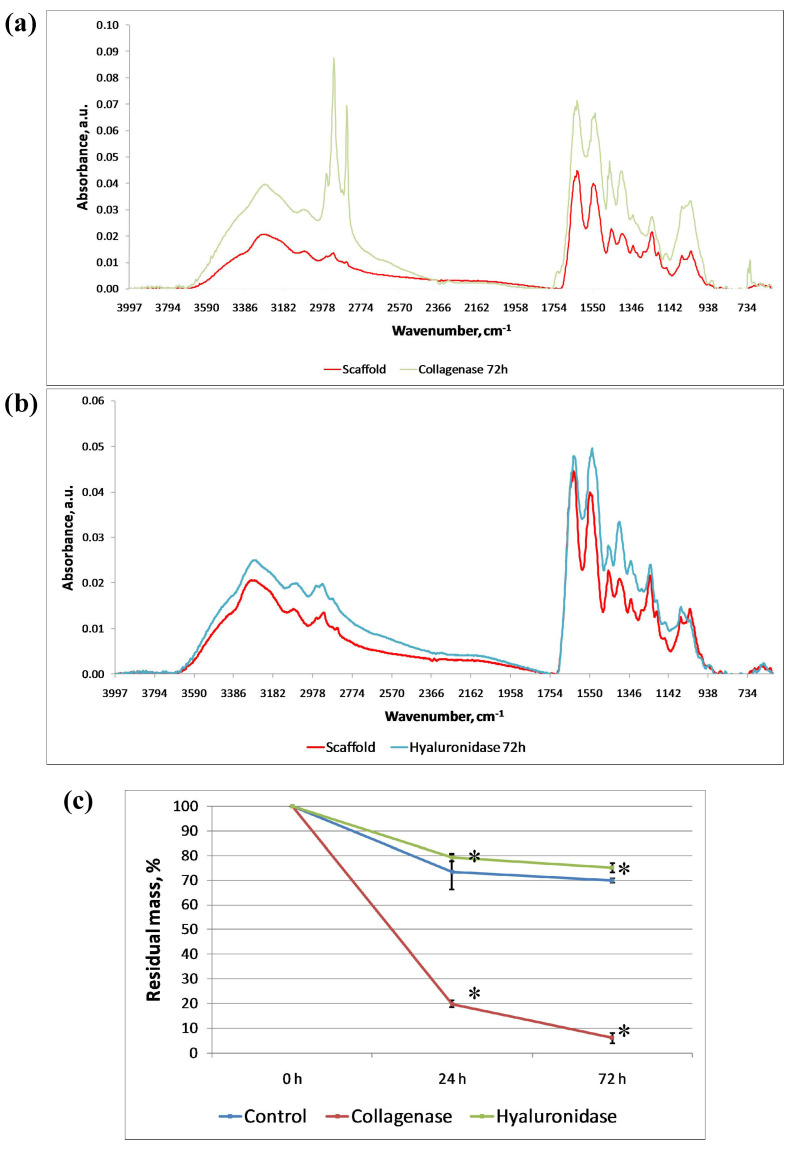
(**a**) Fourier transform infrared (FTIR) spectrum data for scaffold without and after treatment with collagenase. (**b**) FTIR spectrum data for scaffold without and after treatment with hyaluronidase. (**c**) Analysis of biodegradation of scaffold in vitro without and after collagenase (72 h) and hyaluronidase (72 h) treatment. The statistical analysis of variance of the means was assessed by ANOVA and Bonferroni posthoc test (*n* = 9, [mean ± SD], * *p* = 0.000001).

**Figure 4 biomimetics-09-00405-f004:**
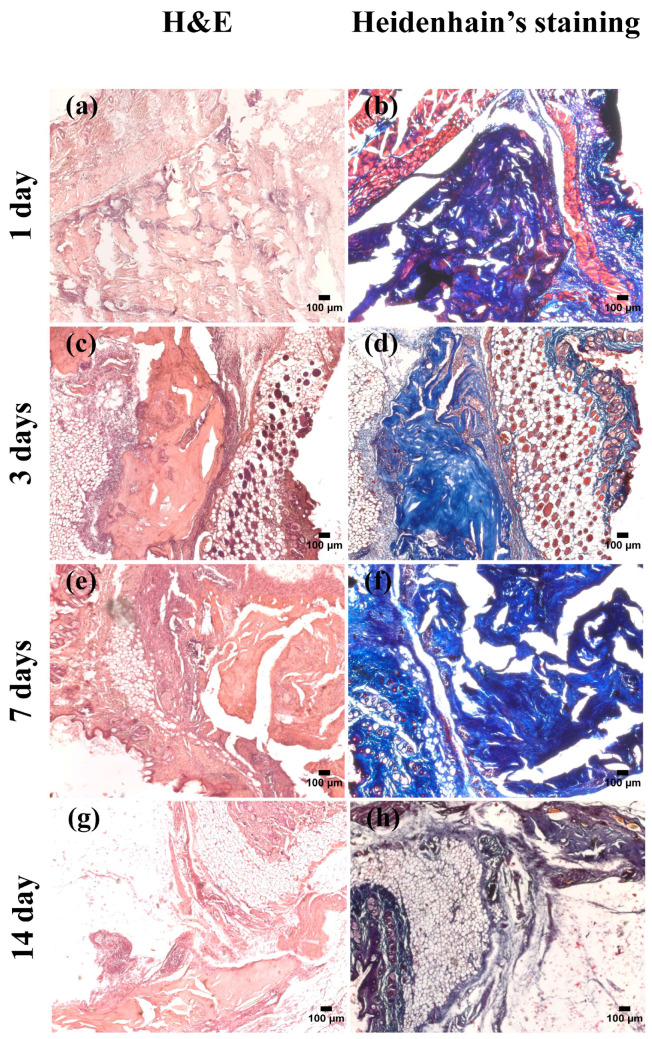
Subcutaneous implantation of scaffold (scale bar = 100 µm): (**a**,**c**,**e**,**g**) H&E staining and (**b**,**d**,**f**,**h**) Heidenhain’s staining.

**Figure 5 biomimetics-09-00405-f005:**
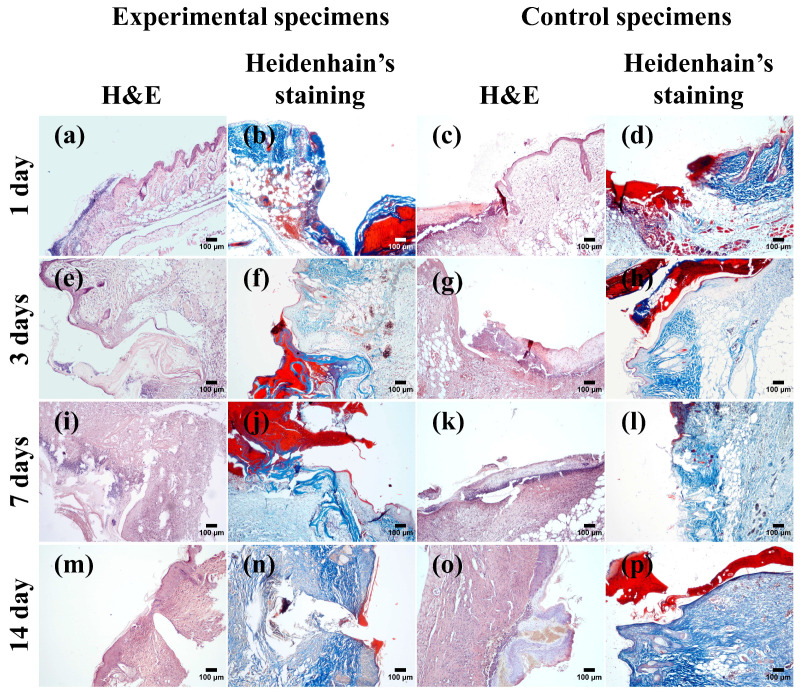
Analysis of mouse full-thickness skin wound healing (scalebar = 100 µm): (**a**,**b**,**e**,**f**,**i**,**j**,**m**,**n**) experimental specimens, (**c**,**d**,**g**,**h**,**k**,**l**,**o**,**p**) control specimens, (**a**,**c**,**e**,**g**,**i**,**k**,**m**,**o**) H&E staining, and (**b**,**d**, **f**,**h**,**j**,**l**,**n**,**p**) Heidenhain’s staining.

**Table 1 biomimetics-09-00405-t001:** Assignments of principal microinfrared vibrational bands of the region of the spectra of the UC-scaffold and Wharton’s jelly.

Infrared Wavenumber Spectra (cm^−1^)				Tentative Assignment	Main Associated Compound
WJ	UC-Scaffold	Incubation in			
		Collagenase	Hyaluronidase		
3278.80	3287.67	3283.64	3271.40	Amide A	Hydrogen bonds, –CH_3_, –CH_2_
2931.74	2929.18	2916.52	2926.72	Amide B	
1633.06	1632.84	1633.66	1631.62	Amide I	Collagen, –C=O
1538.20	1538.96	1539.84	1537.80	Amide II	Collagen, –NH_2_
1453.78	1453.02	1462.34	1454.18	-	–CH_3_
1337.09	1336.75	1337.93	1337.93	Amide III	Collagen
1234.95	1235.51	1237.99	1235.95	Amide III	Collagen, GAGs
1203.86	1203.06	-	1203.32	Amide III	Collagen
1158.94	1160.91	1162.53	1160.49	-	Collagen (carbohydrate moiety), GAGs
1077.12	1079.42	1078.91	1078.91	-	Collagen, sulfated GAGs
1030.70	1031.42	1034.04	-	-	Collagen, GAGs

Key: WJ—Wharton’s jelly; GAGs—glycosaminoglycans.

## Data Availability

The original contributions presented in the study are included in the article; further inquiries can be directed to the corresponding author.

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
