# Peer review of "Decellularized Umbilical Cord as a Scaffold to Support Healing of Full-Thickness Wounds"

_biomimetics, 2024, doi:10.3390/biomimetics9070405_

Round 1

Reviewer 1 Report

Comments and Suggestions for Authors

1.      The research showed a comprehensive analysis of decellularized scaffold derived from human umbilical cord and its efficiency for the wound healing process. Advanced technologies are used to characterized the scaffold which are well justified in the discussion part. However, in the in vitro part of the methodology, the concentration or quantity of scaffold used is not defined. Also, how did the researcher use lyophilized scaffold either soak it before use or directly place it in the cell culture plate? What was the cell count used in the Dead/Live assay?

2.      In a sentence, “Morphology analysis was conducted using electron microscopy and histological analysis;”, it is suggested to write “Morphological analysis” (line # 18).

3.      In a sentence, “Wound healing is a complicated process comprising hemostasis, inflammatory re-actions, cell proliferation, and tissue remodeling.”, correct the word re-actions to reactions. (line # 36).

4.      In a sentence, “Autologous skin grafts for the treatment of full-layered wounds are limited by tissue availability and existing other methods are only moderately clinically effective, which justifies the need to develop new approaches to the treatment of skin injuries”, it is suggested to write “limited because of/due to tissue availability” instead of “limited by tissue availability”, “other existing methods” instead of “existing other methods” and “for the treatment of skin injuries” instead of “to the treatment of skin injuries” (line # 38-41)

5.      Rephrase the sentence, “One of the most promising biomimetics of ECM, due to the presence in its composition of components involved in the regulation of local tissue homeostasis and the transmission of cellular signals, is decellularized tissue”, for better understanding. (line # 49-51)

6.      In a sentence, “The high biostimulating activity of decellularized tissues made it possible to create a number of cellular and tissue medical products for the treatment of pathological conditions of the skin, cartilage, kidneys, liver, pancreas, lungs and other tissues”, as the tissue is itself decellularized how would it be used to create or develop cellular or tissue medical product? Justify it or use the appropriate alternative. (line # 52-54)

7.      In a sentence, “In addition, the composition of the allogeneic biomaterial is significantly affected by external and internal factors throughout the life of the donor”, which type of factors? It is better to mention few like “such as…” (line # 58-59)

8.      There is space from line#62-64. Is it given intentionally?

9.      In a sentence, “Positively, umbilical cord ECM has immunomodulatory and some bacteriostatic effect”, it is better to write “umbilical cord ECM has positive effects…. etc.” (line # 78-79)

10.  In a sentence, “In vivo experiments, despite the inability to fully reproduce the mechanism of human skin damage and the conditions of its healing, remain an indispensable stage in the research and development of new wound coatings”, please check the term in vivo. Does not it include human as well as other animals? or write in vivo animal model instead. (line # 90-91)

1.  In the sentence, “The preservation of GAG after decellularization was confirmed by histochemical staining with alcyan blue...”, there is spelling mistake for “alcian blue” (line # 384)

Comments on the Quality of English Language

nil

Author Response

COVER LETTER #1

Authors express gratitude to the reviewer for a close reading of the manuscript, comments and recommendations.

To the extent possible, we tried to respond to all the questions and introduce appropriate amendments to the text.

Questions

  1. Also, how did the researcher use lyophilized scaffold either soak it before use or directly place it in the cell culture plate? What was the cell count used in the Dead/Live assay?

Answer 1.

Samples consisted of 5 x 105 hADSCs and 10 mg of the UC-scaffold. UC-scaffolds pre-soaked during the 24 hours in the culture medium were seeded with cells in tubes using orbital shaker Multi Bio 3D (Biosan, Latvia).

This information has been added to the MATERIALS AND METHODS.

  1. In the sentence, “Morphology analysis was conducted using electron microscopy and histological analysis;”, it is suggested to write “Morphological analysis” (line # 18).

Answer 2.

We agree with the reviewer and changed “Morphology analysis was conducted using electron microscopy and histological analysis” to “Morphological analysis”.

  1. In a sentence, “Wound healing is a complicated process comprising hemostasis, inflammatory re-actions, cell proliferation, and tissue remodeling.”, correct the word re-actions to reactions. (line # 36).

Answer 3.

We agree with the reviewer and changed “re-actions” to “reactions”.

  1. In a sentence, “Autologous skin grafts for the treatment of full-layered wounds are limited by tissue availability and existing other methods are only moderately clinically effective, which justifies the need to develop new approaches to the treatment of skin injuries”, it is suggested to write “limited because of/due to tissue availability” instead of “limited by tissue availability”, “other existing methods”instead of “existing other methods” and “for the treatment of skin injuries” instead of “to the treatment of skin injuries” (line # 38-41).

Answer 4.

We agree with the reviewer and changed “Autologous skin grafts for the treatment of full-layered wounds are limited by tissue availability and existing other methods are only moderately clinically effective, which justifies the need to develop new approaches to the treatment of skin injuries”, it is suggested to write” to “Autologous skin grafts for the treatment of full-thickness wounds are limited due to tissue availability, and other existing methods have only moderate clinical efficacy, justifying the need for the development of new approaches for the treatment of skin lesions”.

  1. Rephrase the sentence, “One of the most promising biomimetics of ECM, due to the presence in its composition of components involved in the regulation of local tissue homeostasis and the transmission of cellular signals, is decellularized tissue”, for better understanding. (line # 49-51).

Answer 5.

We agree with the reviewer and changed “One of the most promising biomimetics of ECM, due to the presence in its composition of components involved in the regulation of local tissue homeostasis and the transmission of cellular signals, is decellularized tissue” to “The effects of using extracellular matrix biomimetics are the formation of a microenvironment for cells that promotes the transmission of cellular signals and is involved in the regulation of local tissue homeostasis. Products made from decellularized tissues meet the declared properties and have shown satisfactory results from their use”.

  1. In a sentence, “The high biostimulating activity of decellularized tissues made it possible to create a number of cellular and tissue medical products for the treatment of pathological conditions of the skin, cartilage, kidneys, liver, pancreas, lungs and other tissues”, as the tissue is itself decellularized how would it be used to create or develop cellular or tissue medical product? Justify it or use the appropriate alternative. (line # 52-54).

Answer 6.

We agree with the reviewer and changed “The high biostimulating activity of decellularized tissues made it possible to create a number of cellular and tissue medical products for the treatment of pathological conditions of the skin, cartilage, kidneys, liver, pancreas, lungs and other tissues” to “The properties of decellularized tissues that promote the regulation of cellular functions and modulation of tissue regeneration processes have been used in the development of tissue-engineered constructs. The use of these tissue-engineered constructs is expected to be used to restore dysfunctional organs, such as skin, cartilage, kidneys, liver, pancreas, lungs and other tissues in experimental studies”.

  1. In a sentence, “In addition, the composition of the allogeneic biomaterial is significantly affected by external and internal factors throughout the life of the donor”, which type of factors? It is better to mention few like “such as…” (line # 58-59).

Answer 7.

We agree with the reviewer and changed “In addition, the composition of the allogeneic biomaterial is significantly affected by external and internal factors throughout the life of the donor” to “In addition, the composition of the allogeneic biomaterial is significantly affected by external and internal factors throughout the life of the donor, such as previous illnesses, developmental defects, and side effects of medications taken”.

  1. There is space from line#62-64. Is it given intentionally?

Answer 8.

We agree with the reviewer. Space has been removed.

  1. In a sentence, “Positively, umbilical cord ECM has immunomodulatory and some bacteriostatic effect”, it is better to write “umbilical cord ECM has positive effects…. etc.” (line # 78-79).

Answer 9.

We agree with the reviewer and changed “Positively, umbilical cord ECM has immunomodulatory and some bacteriostatic effect” to “umbilical cord ECM has positive effects”.

  1. In a sentence, “In vivo experiments, despite the inability to fully reproduce the mechanism of human skin damage and the conditions of its healing, remain an indispensable stage in the research and development of new wound coatings”, please check the term in vivo.Does not it include human as well as other animals? or write in vivo animal model (line # 90-91).

Answer 10.

We agree with the reviewer and changed “In vivo experiments, despite the inability to fully reproduce the mechanism of human skin damage and the conditions of its healing, remain an indispensable stage in the research and development of new wound coatings” to “Experiments on in vivo animal models, despite the inability to fully reproduce the mechanism of human skin damage and the conditions of its healing, remain an indispensable stage in the research and development of new wound coatings”.

Thank you very much! The text has been corrected.

  1. In the sentence, “The preservation of GAG after decellularization was confirmed by histochemical staining with alcyan blue...”, there is spelling mistake for “alcian blue” (line # 384).

Answer 11.

We agree with the reviewer and changed “alcyan blue” to “alcian blue”.

Reviewer 2 Report

Comments and Suggestions for Authors

The paper is  well-written and describes a simple method for decellularization of umbilical cord matrix. The article definitely presents the advantages of using this biocompatible, biodegradable matrix, however, the disadvantages are not mentioned anywhere.

Some additional comments for the authors - 

1) For the haDSCs culture, it is unclear how many cells are seeded per scaffold and what the size of the scaffold is (Section 2.10)? 

2) It is also unclear why chondrogenic differentiation was performed and how it was done (Section 2.10)? 

3) For the In vivo study,  0.01gm of scaffold was implanted (section 2.11), however, the details on the size and shape of this matrix are missing. In addition, how does the scaffold fit in the 5mm long wound or the 1.2cm wounds?  

4) What was the scaffold's tensile strength and how were uniform scaffolds prepared? 

Author Response

COVER LETTER #2

Authors express gratitude to the reviewer for a close reading of the manuscript, comments and recommendations.

To the extent possible, we tried to respond to all the questions and introduce appropriate amendments to the text.

Questions

  1. The paper is  well-written and describes a simple method for decellularization of umbilical cord matrix. The article definitely presents the advantages of using this biocompatible, biodegradable matrix, however, the disadvantages are not mentioned anywhere.

Answer 1.

We added “For many applications in regenerative medicine, it will be necessary to prolong the biodegradation time, for this it is advisable to try out various innovations for UC-scaffold lysate crosslinking”

  1. For the haDSCs culture, it is unclear how many cells are seeded per scaffold and what the size of the scaffold is (Section 2.10)? 

Answer 2.

Samples consisted of 5 x 105 hADSCs and 10 mg of the UC-scaffold. UC-scaffolds pre-soaked during the 24 hours in the culture medium were seeded with cells in tubes using orbital shaker Multi Bio 3D (Biosan, Latvia).

This information has been added to the MATERIALS AND METHODS.

  1. It is also unclear why chondrogenic differentiation was performed and how it was done (Section 2.10)? 

Answer 3.

We agree with the reviewer and removed information about chondrogenic differentiation.

  1. For the In vivo study,  0.01gm of scaffold was implanted (section 2.11), however, the details on the size and shape of this matrix are missing. In addition, how does the scaffold fit in the 5mm long wound or the 1.2cm wounds?

Answer 4.

The UC-scaffold is prepared by drying a mass of decellularized Wharton's jelly. It is a porous mass without a definite shape. The UC-scaffold sample was cut using scissors to cover the surface of experimental wounds in mice.

What was the scaffold's tensile strength and how were uniform scaffolds prepared?

Answer 4.

We agree that scaffold's tensile strength is necessary to estimate the mechanical parameters of the UC-scaffold. In the future, we plan to estimate tensile strength of the UC-scaffold while using different cross-linkers to UC-scaffold lysates. When preparing the Wharton scaffold, the jelly was subjected to homogenization, followed by decellularization and lyophilization. The resulting UC-scaffold has a fairly uniform porous structure, which is presented on figure 2.
